# Synergistic Effect of Cold Treatment Combined with Ethyl Formate Fumigation against *Drosophila suzukii* (Diptera: Drosophilidae)

**DOI:** 10.3390/insects13080664

**Published:** 2022-07-22

**Authors:** Jong-Chan Jeon, Hyun-Kyung Kim, Hyun-Na Koo, Bong-Su Kim, Jeong-Oh Yang, Gil-Hah Kim

**Affiliations:** 1Department of Plant Medicine, College of Agriculture, Life and Environment Science, Chungbuk National University, Cheongju 28644, Korea; jchan7475@naver.com (J.-C.J.); nshk0917@gmail.com (H.-K.K.); hyunnakoo@hanmail.net (H.-N.K.); 2Plant Quarantine Technology Center, Animal and Plant Quarantine Agency, Gimcheon 39660, Korea; bskim79@korea.kr (B.-S.K.); joyang12@korea.kr (J.-O.Y.)

**Keywords:** spotted wing drosophila, *Drosophila suzukii*, postharvest quarantine treatment, fumigation, cold treatment, ethyl formate

## Abstract

**Simple Summary:**

A synergistic effect on *Drosophila suzukii* control was observed by combining EF fumigation with cold-temperature treatment. *D. suzukii* showed higher mortality at 1 °C after exposure to cold temperature. The egg stage showed the highest tolerance in the ethyl formate fumigation-only treatment according to the LCT_99_ value in a 12 L desiccator. Among the combination treatment methods, cold treatment after EF fumigation was found to be the most effective for *D. suzukii* control, and mortality increased as the duration of exposure to cold temperature increased. Although the sorption of EF was very high, the concentration of fumigant during treatment had a significant effect on insecticidal activity during combination treatment. Therefore, the combination of EF fumigation and cold-temperature treatment can be used to control *D. suzukii*.

**Abstract:**

*Drosophila suzukii* is a quarantine pest that is rapidly spreading in berries. This study evaluated the synergistic effect of combination treatment with ethyl formate (EF) and cold temperature for *D. suzukii* control on imported grapes. A higher insecticidal effect was observed at 1 °C than at 5 °C at all developmental stages, and the pupal stage showed the strongest tolerance to cold temperature. After EF fumigation alone, eggs showed the highest tolerance at 216.67 mg·h/L (LCT_99_ value), and adults showed the highest susceptibility at <27.24 mg·h/L. Among the combination treatment methods, cold temperature after fumigation resulted in the best synergistic effect. The effect of this combination was significant, with 23.3% higher mortality for eggs, 22.4% for larvae, and 23.4% for pupae than observed with EF fumigation alone. Furthermore, the period of complete *D. suzukii* control in the 12 L desiccator was shorter in the combination treatment group at the LCT_80_ value than at the LCT_50_ value of the egg stage. EF showed a very high sorption rate (24%) after 4 h of exposure at a grape loading ratio of 15% in a 0.65 m^3^ fumigation chamber. As the grape loading ratio for combination treatment decreased, *D. suzukii* mortality increased, but when EF was administered at the LCT_80_ value, there was little difference in the mortalities of the eggs and larvae but not the pupae. All *D. suzukii* developmental stages were completely controlled within 7 days after combination treatment, and phytotoxicity was not observed in grapes. These results suggest that the combination of cold-temperature treatment and EF fumigation could be used for *D. suzukii* control.

## 1. Introduction

Spotted-wing drosophila (*Drosophila suzukii*) originated in East Asia but is currently found on all continents [1]. However, postharvest treatments can help reduce the spread of the pest within continents and in countries where it does not yet occur [2,3]. *D. suzukii* lay eggs on fruits, and the hatching larvae burrow into the fruit, making the detection of infection difficult at the early stage. This also causes serious economic losses due to the secondary damage caused by other insects and pathogens from spoilage [3,4,5,6,7].

In general, quarantine disinfection to control *D. suzukii* is performed chemically by using fumigants such as methyl bromide (MB), phosphine (PH_3_), and ethyl formate (EF) or by various physical controls such as heat treatment, low-temperature treatment, and irradiation treatment [2,8,9,10].

EF is a substance naturally volatilized by plants that is used to control pests in stored grains, dried fruits, raw fruits, and wood in Australia and in exported agricultural products such as bananas, oranges, and pineapples in the Philippines [11,12,13,14,15]. EF is rapidly hydrolyzed, has no risk of bioaccumulation or genotoxicity, and is designated as a safe substance by the US Food and Drug Administration (FDA) that appears on the list of substances generally recognized as a safe (GRAS) [12,16,17,18,19]. However, its high cost and possible damage to some crops are issues that need to be solved [20].

The physical control of pests using temperature is widely used internationally, and cold-temperature treatment is being actively introduced in Japan [21,22,23,24]. It has been reported that cold-temperature treatment for the control of quarantine pests of the order Dipteran, such as the Caribbean fruit fly *Anastrepha suspensa* (Loew) and the oriental fruit fly *Bactrocera dorsalis* (Hendel), is effective [25,26].

Recently, pest control using combination treatment rather than a single treatment with fumigants or physical control methods has been researched. Various fumigant treatments at low temperatures have been reported for many pests, such as the Mediterranean fruit fly *Ceratitis capitata* (Dipteran: Tephritidae), *B. dorsalis*, the peach fruit moth *Carposina niponensis* (Lepidoptera: Carposinidae), and the yellow peach moth *Conogethes punctiferalis* (Lepidoptera: Crambidae) using MB; *C. niponensis*, *B. dorsalis,* and the guava fruit fly *Bactrocera correcta* using PH_3_; and *D. suzukii* and the mushroom fly *Lycoriella mali*, using EF [22,27,28,29,30,31,32].

In this study, the synergistic insecticidal effects of EF fumigation and cold treatment were researched to assess their potential combined use as a *D. suzukii* control method in grapes.

## 2. Materials and Methods

### 2.1. Insect

*D. suzukii* was reared using artificial food and sugar solution (20%) in the Insect Toxicology Laboratory of Chungbuk National University with support from Gyeonggi-do Agricultural Research and Extension Services, Republic of Korea, in 2020. The artificial food for rearing egg, larval, and pupal stages was modified using the method of Dalton et al. [33] and boiled for 15 min by mixing with distilled water (2 L), agar (16 g), cornmeal (168 g), sugar (75.2 g), dry yeast (48 g), methyl paraben (3.2 g, Samchun Chemicals, Pyeongtaek, Korea), and green food coloring (2 mL, Saerohands, Namyangju, Korea). When the temperature dropped to 63 °C, propionic acid (22.8 mL, Samchun Chemicals, Pyeongtaek, Korea) was added, and the mixture was placed in a breeding dish (100 mm i.d.). *D. suzukii* adults were provided sugar solution (20%), and breeding dishes with artificial food were changed daily for offspring.

The rearing conditions consisted of incubation at 20 ± 1 °C and 60 ± 10% relative humidity under a 16:8 h light:dark cycle.

### 2.2. Fumigant and Crop

EF (97%) was purchased from Sigma–Aldrich (St. Louis, MO, USA), and the crop used in the experiment was the grape cultivar ‘Campbell Early’ harvested from a domestic vineyard in 2021.

### 2.3. Fumigation Experiments

The fumigation activities of EF against *D. suzukii* were observed in a 12 L desiccator (Duran, Mainz, Germany) for 4 h of exposure following the method of Cho et al. (2020) with modification [34].

Various amounts of EF were applied to filter paper (90 mm i.d.) using a 100 µL gastight syringe (Hamilton, NV, USA) inside a 12 L desiccator. *D. suzukii* eggs and larvae (30 of each) were positioned in a Petri dish (100 mm i.d.) containing artificial food. Thirty pupae were set on filter paper soaked in water and placed in a Petri dish, and adults were placed in only a Petri dish for the experiment. The eggs were observed until pupation from the egg laid within 6 h, and then the number of live pupae was recorded after 12 d. The experiment was conducted using third-instar larvae, pupae within 2 d after pupation, and adults within 3 d after emergence. The mortality of the pupae after 7 d, emergent adults after 9 d, and adults after 1 d was observed.

All experiments were repeated at least 3 times, and the control was not treated with any fumigant. The experimental conditions were maintained at 20 ± 1 °C and 60 ± 10% relative humidity under a 16:8 h light:dark cycle.

### 2.4. Cold Treatment Experiments

Mortality was measured according to cold treatment (1 °C and 5 °C) exposure time (ET) for all developmental stages of *D. suzukii* using a cold chamber (JS Research INC., Gongju, Korea).

All developmental stages of *D. suzukii* were placed in Petri dishes in the same way as described above for the fumigation experiments and treated at cold temperatures. Cold treatments were carried out for 7 d for each developmental stage, and mortality was investigated after 14 d for the egg stage, 9 d for the larval stage, 12 d for the pupal stage, and 2 d for the adult stage.

All experiments were performed at least 3 times, and the control was performed without any cold treatment at 20 ± 1 °C.

### 2.5. Combination Treatment Experiments

*D. suzukii* mortality was investigated by combining EF and cold treatment at 1 °C using a 12 L desiccator and a 0.65 m^3^ fumigation chamber (230 × 50 × 50 cm).

Insecticidal activity against *D. suzukii* was determined using 2 combination treatment methods: fumigation treatment followed by cold treatment and fumigation after cold treatment. EF involved exposure to the LCT_50_ value at the egg stage for 4 h at 20 ± 1 °C in a 12 L desiccator, and cold treatment (1 °C) was carried out for 1 d followed by incubation at 20 ± 1 °C.

Next, the cold treatment exposure time was increased for 7 d to observe *D. suzukii* mortality using the cold treatment experimental protocol after fumigation, which showed the most effective combination activity in a 12 L desiccator. EF treatment was carried out for 4 h at the LCT_50_ and LCT_80_ values for the egg stage determined using single treatments, and cold treatment alone was performed for 7 d.

Another *D. suzukii* mortality experiment was performed by applying EF fumigation for 4 h with 10% and 15% loading ratios (*w*/*v*) of grapes in a 0.65 m^3^ fumigation chamber. EF was treated with LCT_50_ and LCT_80_ values at the egg stage for 4 h at 20 ± 1 °C, and cold treatment (1 °C) was carried out for 7 d to confirm the synergistic effect.

The mortality at each developmental stage, with the exception of the adult stage, was determined in the same way as in the cold treatment experiment. All experiments were performed with at least 3 replicates of 30 insects for each developmental stage.

### 2.6. Gas Concentration and Sorption Measurements

The concentration and time (CT) values were calculated by collecting gases at 0.5 h, 1 h, 2 h, and 4 h after EF treatment [35]. A total of 50 mL of EF gas in each 12 L desiccator was collected in a Tedlar gas sampling bag (1 L, SKC, Dorset, UK) using a syringe (100 mL, Hamilton, NV, USA). The EF concentrations were analyzed using gas chromatography (GC; Agilent Technology 6890N, Agilent Technology, Santa Clara, CA, USA) with the following conditions: flame ionization detector (FID) injector temperature of 200 °C, oven temperature of 100 °C, and detector temperature of 240 °C while utilizing an HP-5 column (0.32 mm × 30 m, Agilent Technology, Santa Clara, CA, USA).

The sorption ratio of EF was determined at grape loading ratios of 0%, 5%, 10%, 15%, and 20% (*w*/*v*) using a 12 L desiccator. All sorption experiments included treatment with 20 mg/L EF at 20 °C for 4 h. The gas concentrations for sorption were determined at 10 min, 30 min, 1 h, 1.5 h, 2 h, 3 h, and 4 h after treatment. C/C_0_ values were calculated as the concentration at each time point after treatment (C) divided by the concentration 10 min after treatment (C_0_). A 12 L desiccator without grapes was used as the control (0%).

### 2.7. Grape Quality Evaluation

Grape quality was used to evaluate the effects of 30 mg/L EF fumigation at 20 °C for 4 h in a 0.65 m^3^ fumigation chamber filled with a 10% grape loading ratio, followed by cold treatment at 1 °C for 24 h of exposure. After combination treatment, fifteen grape clusters were randomly collected on days 3, 7, 10, and 14 d of storage at 5 °C to evaluate quality. As a control, grapes stored at 5 °C without any treatment were assessed. Surface color, sugar content, weight loss, decay rate, and berry abscission were observed for quality evaluation. The surface color was examined for brightness (L), redness (a), and yellowness (b) using a chromameter (CR-400, Minolta Inc., Osaka, Japan). The sugar content was measured using a refractometer (Atago Co. Ltd., Tokyo, Japan). The weight loss rate was determined as a percentage using the weight of each grape cluster. The rate of decay was calculated by dividing the number of decayed grape berries by the total number of grape berries. The berry abscission rate was ascertained by placing the grapes in a shaker (N-Biotec Inc., Bucheon, Korea) at 150 rpm for 1 min and then calculating the number of berries that had undergone abscission as a percentage.

### 2.8. Statistical Analysis

The lethal concentration and time (LCT) values after a single EF fumigation treatment and the lethal exposure time (LET) values of the cold-temperature treatments (1 °C and 5 °C) alone were calculated using probit analysis for all developmental stages of *D. suzukii* [36]. The mortality of all *D. suzukii* at each developmental stage (except adults) under both combination treatments using EF fumigation and cold temperature (1 °C) and the mortality according to the EF LCT values (LCT_50_ and LCT_80_ values of egg stage) and grape loading ratio were compared and analyzed using *t*-tests [36]. The differences in grape phytotoxicity between the combination treatment and control were also analyzed using a *t*-test [36].

## 3. Results

### 3.1. Effects of EF Fumigation on D. suzukii

The effects of EF fumigation on all *D. suzukii* developmental stages were investigated in a 12 L desiccator (Table 1).

Comparing the LCT values at each *D. suzukii* developmental stage showed that adults had the highest susceptibility and 100% fumigation activity with <27.24 mg·h/L EF. The egg, larval, and pupal stages showed similar activities, and in particular, the tolerance ratio (TR) values were 7.95, 7.22, and 7.34 times higher than the LCT_99_ value at the adult stage, respectively. *D. suzukii* eggs showed the highest tolerance to EF fumigation.

### 3.2. Effects of Cold Treatment on D. suzukii

The effect of cold temperature (1 °C and 5 °C) exposure time on mortality was investigated for all developmental stages of *D. suzukii* (Table 2).

When comparing the LET values of all developmental stages of *D. suzukii*, a longer LET was observed at 5 °C than at 1 °C, showing susceptibility to low temperature. There were no significant differences in the cold temperature TR values at any developmental stages, but the longest LET value was found in adults. Additionally, the eggs were the most susceptible to both cold-temperature treatments.

### 3.3. Effects of the Combination of Fumigation and Cold Treatment

The insecticidal activity against all *D. suzukii* developmental stages except adults was investigated with two combination treatments consisting of fumigation for 4 h and cold temperature (1 °C) exposure for 24 h (Figure 1).

Both combination treatments resulted in higher *D. suzukii* mortality at all developmental stages than fumigation alone according to the LCT_50_ value of the egg stage. A greater insecticidal effect was observed at 21.0% for eggs, 14.0% for larvae, and 10.5% for pupae that received cold treatment after EF fumigation than for cold treatment followed by fumigation. Cold treatment after fumigation differed significantly from treatment with fumigation alone.

After exposure to fumigation followed by cold treatment (1 °C for 7 d), *D. suzukii* mortality was compared by the time of exposure using the LCT_50_ and LCT_80_ values of the egg stage (Figure 2).

A very low ovicidal effect (21.8%) was observed after 1 d of cold treatment alone, but a large synergistic effect was found with the combination treatment. The most susceptible stage, the larval stage, showed 100% mortality after EF fumigation treatment at the LCT_50_ and LCT_80_ values of the egg stage for 4 h and exposure to cold treatment for 2 d and 1 d. The pupal stage showed the greatest tolerance to the combination treatment, and 100% mortality was observed after 5 d of cold treatment following EF fumigation at the LCT_50_ value.

### 3.4. Sorption of EF on Grapes

The sorption ratios of EF with various loading ratios of grapes in a 12 L desiccator were investigated (Figure 3).

The EF C/C_0_ values decreased by 16% after 4 h in the control group without grapes, but the EF concentration decreased sharply when the grape loading ratio was over 10%. The EF C/C_0_ values decreased to 65% and 57% at 10% and 15% grape loading ratios, respectively, after 2 h of fumigation treatment, but this value decreased dramatically to 26% after 2 h at a grape loading ratio of 20%.

### 3.5. Effectiveness of Combination Treatment According to the Grape Loading Ratio

The effectiveness of each fumigant concentration (LCT_50_ and LCT_80_ values during the egg stage) at different grape loading ratios (10% and 15%) against *D. suzukii* in combination with cold treatment was investigated in a 0.65 m^3^ fumigation chamber according to cold treatment exposure time (Figure 4).

In eggs, both the LCT_50_ and LCT_80_ values of EF followed by treatment at 1 °C resulted in similar mortality regardless of the grape loading ratio. However, the larval and pupal stages showed significant differences in mortality according to the grape loading ratio at the LCT_50_ value of the egg stage. There was no significant difference in the loading ratio when fumigation was performed at the LCT_80_ value during combination treatment at all developmental stages except the adult stage. When the egg and larval stages of *D. suzukii* were treated with the LCT_80_ of EF at 1 °C for 1 d, over 93% toxicity was observed.

### 3.6. Grape Quality Changes

The effect of cold temperature (1 °C) after fumigation (30 mg/L) on grapes was investigated (Table 3).

There was no significant difference in the grapes between the combination treatment group and the control, although the weight loss and decay rate increased over time. The berry abscission rate increased after combination treatment but was not significantly different from that in the control. The sugar content in the grapes was not related to time after treatment, and there was also no significant difference from the control. Regarding the change in surface color, the brightness value (L) of the treated grapes was lower, and the yellowness value (b) was higher than those of the control grapes, but no statistically significant difference was observed.

## 4. Discussion

There are existing studies on different postharvest treatments, but in this study, the effect of combination treatment using cold temperature, a physical control method, was tested to increase the efficiency of *D. suzukii* control using EF [2]. The effects of EF differ according to the developmental stage of many insects [32,37,38,39]. *D. suzukii* eggs showed the highest tolerance to EF fumigation, with an LCT_99_ value of 168.5 mg·h/L when treated at 21 °C for 4 h, while adults were the most susceptible, with an LCT_99_ value of ≤5 mg·h/L [32]. Similar results were found in this study, as eggs showed higher tolerance to EF fumigation than adults. *Tetranychus urticae* (Trombidifores: Tetranychidae) eggs showed higher EF tolerance than adults, while *Phthorimaea opercullella* (Lepidoptera: Gelechiidae) pupae showed the highest tolerance, with the adults being susceptible [37,40]. The nymph stages of two species of mealybugs (*Pseudococcus longispinus* and *Pseudococcus orchidicola* (Hemiptera: Pseudococcidae) have been shown to have greater tolerance to EF than adults [38]. It was also found that fumigation activity could be reduced in the low-respiration egg and pupal stages due to the characteristics of the fumigant, which is highly related to respiration [41,42].

When cold temperature was used as a physical control method, the stored product pests *Sitophilus granarius* (Coleoptera: Curculionidae), *Callosobruchus rodesianus* (Coleoptera: Bruchidae), *Ephestia cautella* (Lepidoptera: Pysalidae), and *Ephestia kuehniella* (Lepidoptera: Pysalidae) showed 100% mortality after 4 h of exposure to a temperature of −18 °C [22]. *B. dorsalis* control is 99% effective after cold-temperature treatment at 5 °C, 6 °C, and 7 °C for 8 d [25,26]. More than 99.9964% of *D. suzukii* can be controlled if treated at 1 °C for 8 d, although in the pupae, which showed the strongest tolerance at 5 °C, cold-temperature treatment for more than 9 d was needed [32,43]. In this study, it was also found that the pupae had the highest cold tolerance under cold treatment and required more than 6 d of exposure at 1 °C for 100% mortality. However, cold temperature alone requires a long treatment time, which is a disadvantage when controlling pests. In general, combination treatment methods, such as applying fumigants in combination with other fumigants or controlling the atmosphere using CO_2_, N_2_, and O_2_ in combination with fumigants, are being studied to reduce disadvantages that may appear with a single treatment [37,44,45]. In addition, combination treatment studies using fumigants and low temperatures, similar to this study, were previously conducted [46,47]. In previous research, the mortality of *Phthorimaea operculella* at all developmental stages except the larval stage at 5 °C and 20 °C did not show significant differences after EF fumigation, and there was no significant difference in the LCT_99_ value even when adult and nymphal *Frankliniella occidentalis* were treated at 5 °C and 10 °C, respectively [40,48]. However, after *T. urticae* were fumigated with EF and treated at 5 °C, 10 °C, and 20 °C, both eggs and adults showed increased susceptibility as the temperature increased [37]. Combining fumigation with low-temperature treatment causes differences in activity depending on insect species and developmental stage, but the fumigation activity of EF seems to have little effect on temperature. This study evaluated the effects of EF and cold temperature on *D. suzukii* by administering the treatments separately, rather than simultaneously, and the results showed that treatment order had a strong effect on *D. Suzukii* mortality.

In general, the sorption of fumigants varies depending on various factors such as the fumigant (PH_3_ and ethanedinitrile), treatment temperature, and products to be treated [49,50]. In particular, EF has a high sorption rate, and the higher the loading ratio was, the higher the sorption rate in this study. When EF and PH_3_ were applied to tobacco leaves, the sorption of EF was high, and the sorption amount was also high in wheat [51,52]. In the previous study, the EF concentration decreased by more than 50% within 2 h at a blueberry loading ratio of 10%, showing similar results to this study [32]. Since EF has a high sorption rate, increasing its concentration is a way to reduce the low-temperature treatment time needed for complete pest control. By adjusting the EF concentration and time at low temperature, this combination treatment method could be an effective strategy for crops that are stored at low temperature by increasing the control effects via short-term low-dose fumigant administration. Furthermore, in the grape phytotoxicity evaluation in this study, EF did not result in any significant difference in quality compared to that of the control until 14 d, and the results of blueberry quality in response to EF fumigation were not significantly different [32].

Therefore, these results suggest that the application of cold-temperature (1 °C) treatment after EF fumigation for 4 h is a strategy that could be used for *D. suzukii* control.

## 5. Conclusions

We studied the combined effects of using cold-temperature treatment (1 °C) for 3 d after EF fumigation for 4 h with the LCT_80_ value and found 100% mortality for all *D. suzukii* developmental stages, except the pupal stage. Thus, this combination treatment could be useful for *D. suzukii* control on grapes stored at cold temperatures for export.

## Figures and Tables

**Figure 1 insects-13-00664-f001:**
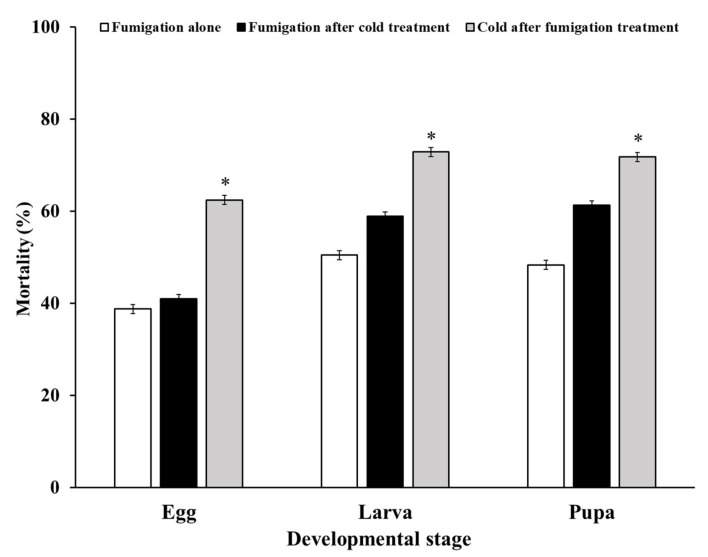
Effects of EF fumigation alone and the two combination treatments on the control of *D. suzukii*. * Indicates a significant difference according to the *t*-test at *p* < 0.05.

**Figure 2 insects-13-00664-f002:**
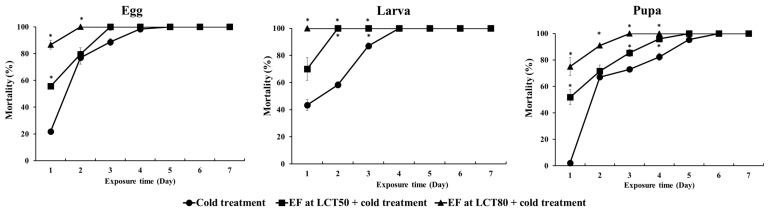
Effects of cold treatment alone and the two combination treatments according to EF concentration (LCT_50_ and LCT_80_ values) and cold-temperature exposure time. * indicates a significant difference according to the *t*-test at *p* < 0.05.

**Figure 3 insects-13-00664-f003:**
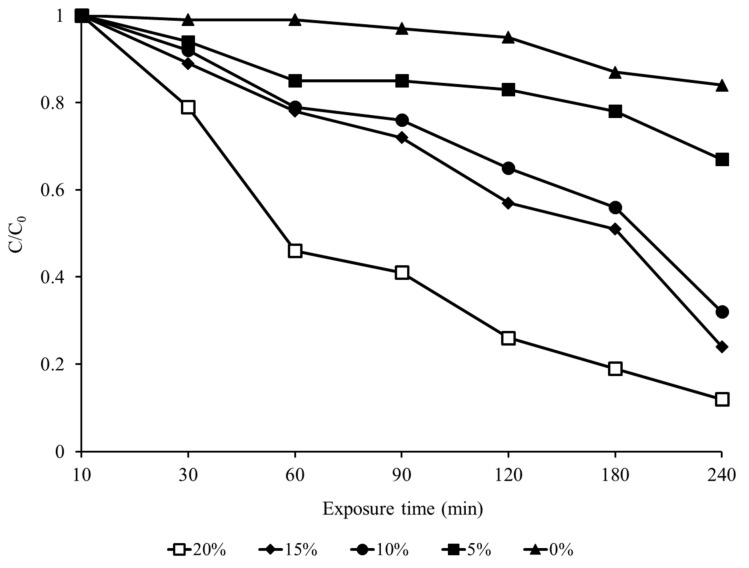
EF sorption concentrations according to different grape loading ratios (0%, 5%, 10%, 15%, and 20%) during fumigation in a 12 L desiccator with 20 mg/L EF for 4 h.

**Figure 4 insects-13-00664-f004:**
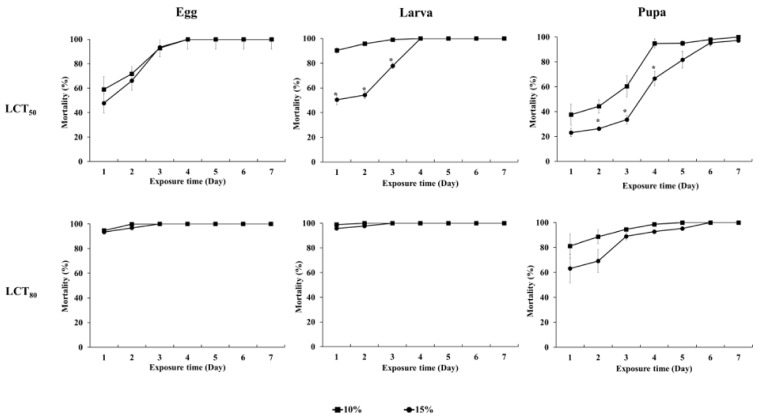
Fumigation effects of combination treatment according to the grape loading ratio (10% and 15%) in a 0.65 m^3^ fumigation chamber. * indicates a significant difference according to the *t*-test at *p* < 0.05.

**Table 1 insects-13-00664-t001:** Toxicity of EF against *Drosophila suzukii* in 12 L desiccator for 4 h of exposure at 20 °C.

Stage	*n*	LCT_50_ ^a^ (mg·h/L) (95% CL ^b^)	TR ^c^	LCT_99_ (mg·h/L) (95% CL)	TR	Slope ± SE	*df*	*x* ^2^
Egg	2460	53.03 (40.91–67.74)	1.95	216.67 (134.64~209.07)	7.95	3.81 ± 0.55	3	47.03
Larva	2490	40.24 (37.14–43.40)	1.48	174.98 (147.10–218.55)	6.42	3.64 ± 0.25	4	215.54
Pupa	3400	47.90 (45.14–50.71)	1.76	199.94 (170.35–245.09)	7.34	3.75 ± 0.24	7	242.68
Adult	2551	<27.24	1	<27.24	1	-	-	-

^a^ LCT_50_ and _99_; 50% and 99% lethal concentration times. ^b^ Confidence limit. ^c^ Tolerance ratio.

**Table 2 insects-13-00664-t002:** Effect of cold temperature (1 °C and 5 °C) exposure time on the control of *Drosophila suzukii*.

Stage	Temp. (°C)	*n*	LET ^a^ _50_ (95% CL ^b^)	TR ^c^	LET_99_ (95% CL)	TR	*df*	Slope ± SE
Egg	1	1170	35.06 (31.87–38.08)	1.12	111.47 (96.58–134.68)	1	3	4.63 ± 0.37
	5	730	45.83 (42.28–49.23)	1.46	135.42 (118.89–160.38)	1.21	3	4.94 ± 0.84
Larva	1	660	31.41 (27.45–35.02)	1	148.17 (121.83–193.39)	1.33	3	3.45 ± 0.88
	5	1434	41.22 (37.42–44.87)	1.31	160.77 (136.04–200.60)	1.44	3	3.94 ± 0.47
Pupa	1	1109	49.13 (45.23–52.85)	1.56	160.46 (141.71–187.70)	1.44	5	4.53 ± 0.76
	5	1350	63.68 (58.85–68.11)	2.03	189.14 (169.74–217.16)	1.70	5	4.9 ± 0.50
Adult	1	1082	58.75 (54.73–62.68)	1.87	184.52 (163.55–214.50)	1.66	5	4.68 ± 0.30
	5	1150	77.08 (72.87–81.16)	2.45	194.77 (175.44–222.47)	1.75	5	5.78 ± 0.39

^a^ Lethal exposure time (h). ^b^ Confidence limit. ^c^ Tolerance ratio.

**Table 3 insects-13-00664-t003:** Phytotoxicity to grapes of the combination of EF and cold treatment in a 0.65 m^3^ fumigation chamber.

DAT ^a^	Treatment	Weight Loss (%)	Berry Abscission (%)	Decay Rate (%)	Sugar Content (%, brix)	Mean Surface Color (Mean ± SE)		
						L	a	b
3	Control	2.1 ± 0.2	0.0 ± 0.0	0.0 ± 0.0	16.1 ± 0.4	105.6 ± 1.2	3.0 ± 0.4	−4.1 ± 1.4
	Combination treatment	2.3 ± 0.1	0.0 ± 0.0	0.0 ± 0.0	16.5 ± 0.3	101.2 ± 1.1	2.6 ± 0.3	−1.8 ± 0.7
	*p* ^b^	0.456	-	-	0.452	0.060	0.433	0.388
7	Control	3.3 ± 0.3	0.0 ± 0.0	0.0 ± 0.0	15.9 ± 0.5	99.1 ± 0.6	1.0 ± 0.5	1.3 ± 0.9
	Combination treatment	4.0 ± 0.9	0.0 ± 0.0	0.0 ± 0.0	16.4 ± 0.3	98.9 ± 1.0	1.1 ± 0.5	1.8 ± 0.5
	*p*	0.558	-	-	0.430	0.918	0.828	0.641
10	Control	5.1 ± 0.6	0.0 ± 0.0	0.0 ± 0.0	16.4 ± 0.6	97.5 ± 0.5	−0.1 ± 0.4	3.6 ± 1.4
	Combination treatment	5.6 ± 0.5	0.5 ± 0.5	1.3 ± 0.7	16.5 ± 1.1	95.6 ± 1.9	−0.2 ± 0.9	6.7 ± 1.4
	*p*	0.511	0.423	0.185	0.939	0.384	0.957	0.182
14	Control	5.2 ± 0.3	0.0 ± 0.0	0.0 ± 0.0	16.6 ± 0.1	97.1 ± 1.3	−6.4 ± 1.0	8.3 ± 1.0
	Combination treatment	6.2 ± 0.5	1.4 ± 0.7	1.6 ± 0.9	16.4 ± 0.3	96.8 ± 1.1	−7.5 ± 0.9	9.4 ± 0.8
	*p*	0.462	0.184	0.209	0.633	0.154	0.472	0.435

^a^ Day after treatment. ^b^ A *t*-test was used to compare the values (%, mean ± SE) of each quality criterion between the control and combination treatments.

## Data Availability

Not applicable.

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
