# Peer review of "Synergistic Effect of Cold Treatment Combined with Ethyl Formate Fumigation against Drosophila suzukii (Diptera: Drosophilidae)"

_insects, 2022, doi:10.3390/insects13080664_

Round 1

Reviewer 1 Report

The manuscript brings relevant information about Drosophila suzukii management. The discussion needs to be improved. I suggest consulting the recent chapter by Walse et al. (2020) as a starting point. Other suggestions are in the attached file. In this item, the authors used many papers on insects with a sizeable phylogenetic distance from D. suzukii, many from other orders. 

Author Response

Reviewer 1

The manuscript brings relevant information about Drosophila suzukii management. The discussion needs to be improved. I suggest consulting the recent chapter by Walse et al. (2020) as a starting point. Other suggestions are in the attached file. In this item, the authors used many papers on insects with a sizeable phylogenetic distance from D. suzukii, many from other orders. 

  • The reviewer’s comments have been taken into consideration as much as possible. Walse et al. (2020) was inserted as a reference. The advice in the attached file also has been revised as much as possible.

Reviewer 2 Report

The manuscript insects-1785376 reports studies to evaluate the efficacy of ethyl formate fumigation in combination with cold temperatures against Drosophila suzukii. The method is intended for quarantine treatment of grapes. Although the paper presents data on one of the promising methyl bromide alternatives, there are several issues with the manuscript, and it is not suitable for publication for the following reasons:

1- The idea of combining ethyl formate with cold storage is not new and has been previously reported in many literatures. For example, Kwon, T. H., Park, C. G., Lee, B. H., Zarders, D. R., Roh, G. H., Kendra, P. E., & Cha, D. H. (2021). Ethyl formate fumigation and ethyl formate plus cold treatment combination as potential phytosanitary quarantine treatments of Drosophila suzukii in blueberries. Journal of Asia-Pacific Entomology, 24(1), 129-135.

2- The manuscript is written in a way that renders it difficult to understand and follow. Many sentences are awkward or unclear, not concise, and need to be rephrased. It needs a lot of proofreading and intense English improvement.

3- The authors claimed to study the synergistic effect of ethyl formate combined with cold treatment. However, they did not use any known formula to determine the synergistic ratios.

4- Many parts in the Materials and Methods Section are not clear/precise and more details are needed. For example:

-          Lines 117 – 123: how do they place adult flies in Petri dishes?  They also mentioned that “eggs were observed until the pupae from the egg laid within 6 h” which cannot be true.

-          Line 156: “10% and 15% loading ratios….” there are different methods to determine the fruit loading ratios. It is very confusing if not mentioning how they calculate it. Is it based on v/v, w/w, w/v, etc.? How the 10% and 15% loading ratios represent those used in commercial setting for grapes?

-           Lines 170 – 171: details are needed on how CT values were determined.

-          Lines 172 – 179: the method used to determine the sorption of ethyl formate in grapes has serious flaws. It is known that EF in the headspace is depleted as a result of many reasons such as degradation, diffusion, sorption, etc. So, the depletion of EF cannot be used to represent its sorption in fruit. There are other accurate methods in literature where researchers analyze the fruit itself.

5- The Discussion Section is a kind of review of the literature and lacks the discussion of the phenomena observed in the Results Section considering relevant references. 

6- The conclusion needs to be rewritten on the basis of sufficient data analysis.

Author Response

Reviewer 2

The manuscript insects-1785376 reports studies to evaluate the efficacy of ethyl formate fumigation in combination with cold temperatures against Drosophila suzukii. The method is intended for quarantine treatment of grapes. Although the paper presents data on one of the promising methyl bromide alternatives, there are several issues with the manuscript, and it is not suitable for publication for the following reasons:

The idea of combining ethyl formate with cold storage is not new and has been previously reported in many literatures. For example, Kwon, T. H., Park, C. G., Lee, B. H., Zarders, D. R., Roh, G. H., Kendra, P. E., & Cha, D. H. (2021). Ethyl formate fumigation and ethyl formate plus cold treatment combination as potential phytosanitary quarantine treatments of Drosophila suzukii in blueberries. Journal of Asia-Pacific Entomology, 24(1), 129-135.

  • Similar but not identical experiment. In order to find a more effective Drosophila suzukii control method, the temperature conditions (5°C and 1°C) and the various loading ratio of grapes were comparatively analyzed and, in particular, an experiment was conducted to scale up even in 0.65 m3 fumigation chamber. An efficient control strategy was indicated by comparing the sorption rate according to the loading ratio of grapes and the concentration of the treated fumigant. In addition, this study includes experiments on all developmental stages in scale up experiment.

It is considered to be a study that distinguishable from previous study.

The manuscript is written in a way that renders it difficult to understand and follow. Many sentences are awkward or unclear, not concise, and need to be rephrased. It needs a lot of proofreading and intense English improvement.

  • I entrusted it to an English proofreading company (AJE) and revised it again.

The authors claimed to study the synergistic effect of ethyl formate combined with cold treatment. However, they did not use any known formula to determine the synergistic ratios.

  • Rather than suggesting a synergistic ratio, this study shows that there is a synergistic effect when combination treatment with low temperature and fumigation. Therefore, it is a study showing that the effect is increased when the temperature is low (1°C) and the fumigation concentration (LC80 value) of the appropriate EF is treated. As a result, this study indicates that low temperature, concentration of fumigant, loading ratio of product, and adsorption of fumigant are interrelated for effective pest control.

Many parts in the Materials and Methods Section are not clear/precise and more details are needed. For example:

Lines 117 – 123: how do they place adult flies in Petri dishes?  They also mentioned that “eggs were observed until the pupae from the egg laid within 6 h” which cannot be true.

  • revised “eggs were observed until pupation from the egg laid within 6 h”

Line 156: “10% and 15% loading ratios….” there are different methods to determine the fruit loading ratios. It is very confusing if not mentioning how they calculate it. Is it based on v/v, w/w, w/v, etc.? How the 10% and 15% loading ratios represent those used in commercial setting for grapes?

  • inserted (w/v). As far as I know, there is nothing definite in Korea. In this study, it was used in the experiment by placing it in a grape box for export as much as possible.

Lines 170 – 171: details are needed on how CT values were determined.

  • inserted and revised.

Lines 172 – 179: the method used to determine the sorption of ethyl formate in grapes has serious flaws. It is known that EF in the headspace is depleted as a result of many reasons such as degradation, diffusion, sorption, etc. So, the depletion of EF cannot be used to represent its sorption in fruit. There are other accurate methods in literature where researchers analyze the fruit itself.

  • This experiment did not observe the adsorption of the fruit, but the difference in the concentration of EF. The term sorption may cause confusion, but it is an expression of a decrease in the concentration of EF. In previous papers, the term sorption was also used, but if you want to revise the term, I will do so.

The Discussion Section is a kind of review of the literature and lacks the discussion of the phenomena observed in the Results Section considering relevant references. 

  • This study is about the combined treatment of low temperature and fumigation for supplement the disadvantages of low temperature treatment and fumigants. In addition, it explains that there is a difference in insecticidal activity according to the developmental stages, and shows that experiments using various factors such as insect species, fumigants, and temperature conditions should be carried out.

I thought I have reviewed to related references, but I have inserted a little more about the lacking parts.

The conclusion needs to be rewritten on the basis of sufficient data analysis.

  • revised

Reviewer 3 Report

Drosophila susukii is a quarantine pest that is increasing its presence in different habitats where its hosts are available. When it is already established, countries need to apply quarantine disinfection, through physical and chemical methods. In this article, the efficacy of the combination of cold temperature and ethyl formate fumigation on all developmental stages of D. susukii is investigated. Overall, the experiments are rigorously planned, executed and analyzed.

Some minor comments and typos by page and line number:

Paragraphs in P4 starting at lines 144, 149 and 155. It should be explained that these are three different experiments. In the current state it is confusing and seems like the same experiment, so the procedure is not clear.

P4 Line170 Line170 the sentence is confusing as other time points are mentioned in Line175

P4 Line 18. Why is grape quality evaluated at 30 mg/L when in previous experiments a concentration of 20 mg/L EF was used?

P Line185 “3, 7, 10 and 14 of storage” add days (d)

Table 1 Check TR of larvae  

P6Line230. What statistical test was used?

P7 Line 247.” 21,0% of eggs, 14,0% of larvae …” “for” instead “of”?

P 14 Line 320-321. Start with “According to a previous study…” . Furthermore, the sorption results, as well as results of grape quality, should be discussed with the results published by Kwon et al. 2021

Check spelling for larvae and pupae in all tables and figures

Author Response

Reviewer 3

Paragraphs in P4 starting at lines 144, 149 and 155. It should be explained that these are three different experiments. In the current state it is confusing and seems like the same experiment, so the procedure is not clear.

  • Revised to be distinguishable.

P4 Line170 Line170 the sentence is confusing as other time points are mentioned in Line175

  • CT value indicate concentration and time, gas concentration indicates for sorption of EF when treated with various grape loading ratio. Revised.

P4 Line 18. Why is grape quality evaluated at 30 mg/L when in previous experiments a concentration of 20 mg/L EF was used?

  • Actually 20 mg/L in 12 L desiccator is LCT50 values of egg which is the highest tolerance stage, but the phytotoxicity of grapes were experimented in a 0.65 m3 fumigation chamber, so in this experiment, the maximum dose of EF (30 mg/L, LCT80 value) was treated.

P Line185 “3, 7, 10 and 14 of storage” add days (d)

  • revised

Table 1 Check TR of larvae  

  • check and revised

P6Line230. What statistical test was used?

  • Revised statistical analysis in materials and methods part.

P7 Line 247.” 21,0% of eggs, 14,0% of larvae …” “for” instead “of”?

  • revised

P 14 Line 320-321. Start with “According to a previous study…” . Furthermore, the sorption results, as well as results of grape quality, should be discussed with the results published by Kwon et al. 2021

  • the paper on sorption ‘Kwon et al. 2021’ is cited and inserted into the discussion part.

Check spelling for larvae and pupae in all tables and figures

Thank you very much.